# Endometriosis and Infertility: Prognostic Value of #Enzian Classification Compared to rASRM and EFI Score

**DOI:** 10.3390/jpm12101623

**Published:** 2022-10-01

**Authors:** Arrigo Fruscalzo, Arnaud Dayer, Ambrogio Pietro Londero, Benedetta Guani, Fathi Khomsi, Jean-Marc Ayoubi, Anis Feki

**Affiliations:** 1Clinic of Obstetrics and Gynaecologz, HFR—Hòpital Fribourgeois, Chemin des Pensionnats 2-6, 1708 Fribourg, Switzerland; 2Faculty of Medicine, University of Münster, Domagkstraße 3, 48149 Münster, Germany; 3Academic Unit of Obstetrics and Gynaecology, Department of Neuroscience, Rehabilitation, Ophthalmology, Genetics, Maternal and Infant Health, University of Genova, 16132 Genova, Italy or; 4Ennergi Research (Non-Profit Organization), 33050 Lestizza, Italy; 5Department of Obstetrics and Gynecology and Reproductive Medicine, Hopital Foch–Faculté de Médecine Paris, 92150 Suresnes, France

**Keywords:** endometriosis, fertility, Enzian classification, endometriosis fertility index (EFI), rASRM, assisted reproductive technology

## Abstract

This study’s objective was to compare the predictive validity of the three most utilized classification scores for endometriosis, #Enzian, EFI, and rASRM, in achieving a spontaneous pregnancy or pregnancy via assisted reproductive technology (ART) after surgery for endometriosis. The monocentric retrospective study was carried out from January 2012 to December 2021 at the gynaecology department of the cantonal hospital of Fribourg. Patients consulting for infertility and operated on for endometriosis with histological confirmation were included. The predictive value of #Enzian, rASRM, and EFI was evaluated and compared concerning the prediction of fertility after surgery, both spontaneous and ART, during the following 12 months. A total of 58 women (mean age 33.1 ± 4.57 years) were included. Overall, 30 women achieved a pregnancy, seven spontaneously. Among all women who achieved a pregnancy, there was a lower prevalence of rASRM stage III–IV (16.67% vs. 39.29%, *p* = 0.054). Women achieving a pregnancy had a significantly higher EFI score than others (*p* < 0.05). No significant differences were observed concerning the #Enzian score. In conclusion, the revised #Enzian score is not correlated with pregnancy achievement; EFI score is the only score significantly associated with the pregnancy outcome in women affected by endometriosis.

## 1. Introduction

Endometriosis is one of the most frequent benign pathologies affecting women of reproductive age. The estimated incidence is currently around 6–10%, a value, however, which remains highly variable and imprecise due to the difficulty of diagnosis as well as its correlation to the type of population considered [1]. It is a potentially very disabling disease, characterized by chronic peri-menstrual pain and subfertility or infertility, as well as distortion of anatomical structures and organs linked to their implantation sites [2]. Since the late 1970s, the international community involved in endometriosis management has been trying to design an appropriate staging system and to impose a common language, able to cover the manifold forms of clinical presentation and standardize comparisons for medical research [3,4].

The first classification was published in 1979 by the American Fertility Society (AFS), then it was revised twice, to revised American Fertility Society (rAFS) in 1985 and to revised American Society of Reproductive Medicin (rASRM) in 1997 [5,6,7]. It has been widely used and remains, in fact, the main classification used for endometriosis [4]. However, it has been shown to have limited predictive value concerning fertility outcome and also clinical presentation, being unsuited for depicting deep infiltrating forms of endometriosis [8]. Thus, the endometriosis fertility index (EFI) was conceived as a prognostic score, suited to better evaluate female fertility. The EFI combines the information given by the rASRM score, the fertility history (maternal age, years of infertility, gestation), and the macroscopic evaluation of the current functional status of the fallopian tubes and ovaries [9]. For its part, the Enzian classification proposes to describe the location and extent of endometriotic lesions infiltrating deep tissue, to overcome the shortcomings of rASRM which only describes the adhesion state of the pelvic organs and peritoneal endometriotic lesions [10]. However, the Enzian classification no longer takes into account superficial endometriosis, nor macroscopic involvement of the peritoneum, fallopian tubes, and ovaries.

Recently, a revised version of the Enzian classification (#Enzian) was proposed, including evaluation of the above-mentioned anatomical structures that are pivotal in female fertility [11]. This was proposed as a universal classification, able to bridge the anatomical changes caused by the disease into a staging system correlating to the disease outcomes. However, to date no clinical data have been presented concerning the prognostic value of #Enzian regarding fertility. This study was designed to explore this gap of knowledge, assessing the association between the stage of endometriosis according to the #Enzian classification and the post-operative pregnancy rate, both spontaneous and after the use of assisted reproductive technologies (ART). Results were also compared to the current gold standards, the rASRM classification and the EFI score.

## 2. Material and Methods

### 2.1. Study Design and Patients Included

This was a retrospective monocentric study conducted at the local university hospital, HFR Fribourg, between January 2012 and December 2020.

The study considered all patients having consulted for infertility and having been operated on by laparoscopy or laparotomy with a histologically confirmed diagnosis of endometriosis. Patients were excluded from the study in case of: non-desire for pregnancy; diagnosis of endometriosis not retained by surgical explorations or histological examination; a lack of information or documentation in the patient’s medical records; refusal of informed consent to participate to the study.

This project has been accepted by the Local Ethic Commission (CER-VD number 2021-00258). An informed consent was obtained for each patient included in the study.

### 2.2. Staging of Endometriosis and Primary Outcome

Our study was based on the patient files established during the specialist consultation for infertility in the gynaecology department of the HFR, as well as on the operating protocol, which preceded the start of treatments. We took some relevant data from these files, grouped into different categories: obstetrical anamnesis, gynaecological anamnesis, endometriosis anamnesis, ART anamnesis, pre- and post-operative therapy, and operating protocol, as well as the outcome of spontaneous pregnancy vs. pregnancy after ART. Through the operating protocol, the scores based on the rASRM, #Enzian, and EFI classifications were retrospectively calculated by one of the authors in order to assess their predictivity for a spontaneous pregnancy or pregnancy by ART. In case of uncertainty, an independent revision process of a second author was provided. A pooling of the scores of the #Enzian compartments in the form of a median of scores was also calculated. The mean of the sum of the rASRM stages was also calculated.

Pregnancy was considered for the main outcome if it was observed within 12 months post-operation for a spontaneous pregnancy, as well as 12 months after the ART method was started for pregnancy by ART.

### 2.3. Statistical Analysis

Data were analysed by R (version 4.2.0), considering a significant *p*-value < 0.05 (two-sided). Pregnancy was achieved in 30 women, and a minimum of 21 women without pregnancy were required to detect a large effect size (0.8) in comparing the proportions of the analysed variables with a power of 80% and significance level of 0.05 (two-sided) [12]. Moreover, a minimum of 26 women without pregnancy were required to detect a large effect size (0.8) in comparing two continuous variables with a non-parametric distribution with a power of 80% and significance level of 0.05 (two-sided) [12]. We presented data as mean (± standard deviation), median (with interquartile range—IQR), or prevalence values. We used the Wilcoxon test to perform univariate analysis in the case of continuous variables. The chi-square test or Fisher’s exact test was used for categorical variables. Missing values were considered missing (NA) in statistical analyses except where specified.

## 3. Results

A total of 58 women were scored during the study period. The mean age was 33.1 years (±4.57); 41.38% (24/58) had a previous pregnancy, and three had at least one previous delivery. Extra-uterine pregnancies were diagnosed in six women; four underwent salpingectomies, and two salpingotomies. Thirty women had a pregnancy during the 1-year follow-up time. Seven of them had a spontaneous conception. Twenty-four conceived through an ART technique, and one of these women also conceived once spontaneously. The pregnancy was evolutive in three out of the seven women (42.85%) who conceived spontaneously, and in 19 out of the 24 patients (79.17%) who conceived through ART. The study population is described in Figure 1.

The majority of cases were found to be classified according to rASRM as stage I (46.55%; 27/58), followed by stage II (25.86%; 15/58), and stage IV (17.24%; 10/58). The median rASRM stage was 2 (IQR 1–3). The median EFI score was 8 (IQR 6–9), and the median sum of Enzian scores was 3 (IQR 1–3); meanwhile, the median of the Enzian score sum considering only peritoneal, ovarian, and tubal endometriosis was 1 (IQR 1–2).

Table 1 shows the scores subdivided according to the achievement of a pregnancy during the follow-up period. Among the group achieving a pregnancy, there was a non-significant lower prevalence of rASRM stage III–IV (16.67% vs. 39.29%, *p* = 0.054). Women achieving pregnancy had a significantly higher EFI score than others (*p* < 0.05). As shown in Table 1, no significant differences were observed according to the Enzian classification. Similar findings were found among women who conceived by ART (Table 2).

Considering only the seven cases in which pregnancy was achieved spontaneously, we found a higher EFI median score in the spontaneous pregnancy group than in other women (7.00, IQR 6.00–8.00 vs. 9.00, IQR 8.00–9.00, *p* < 0.05) (Table 3). Furthermore, in these seven women, there was a higher prevalence of endometriosis in the F compartment (42.86% (3/7) vs. 10.71% (5/51), *p* = 0.079). These three cases were located: one in the diaphragm, one in the cervix, and one in the ureter. No other differences were observed in this sub-analysis.

## 4. Discussion

According to our results, the #Enzian classification did not show any prognostic association with subsequent female fertility, either for cumulative pregnancies, or for pregnancies after ART or spontaneous ones. The rASRM classification also did not show any prognostic correlation to later fertility. Conversely, the EFI index showed a significant association with pregnancy outcome, for both assisted reproduction and spontaneous pregnancies.

In terms of pregnancies achieved, our results show that seven spontaneous pregnancies were obtained in the 58 patients (12.07%). A systematic review dating from 2020 combines three different studies, and evaluates in particular the number of women with a pregnancy confirmed by ultrasound at 9 months or 18 months following laparoscopy with ablation and excision of the endometriosis [13]. The spontaneous pregnancy rate was, respectively, 35%, 36.62%, and 12.33%. Various factors may explain our slightly lower result: in the three studies described in this systematic review, one of the inclusion criteria was rASRM stage 1 or 2 (minimal to moderate) whereas, in our study, we also included medium to severe stages (27.59% of all cases). In addition, the rather high average age of the patients in our study (33.1 ± 4.57 years) encouraged proactive management, thus implying management by ART directly in the first months following the operation. It should also be noted that almost 2/3 of the patients had a mixed gynaecological involvement, and not exclusively an endometriotic one, also justifying the proactive attitude; however, tubal permeability (17/58) can be due to multiple causes, in particular to endometriosis. Among the patients who had recourse to ART, 50% (24/48) obtained a pregnancy within 12 months after the start of treatment. Also, we noticed that there was a lower rate of live birth in the group of spontaneous pregnancies compared to the ART group. These results are difficult to compare with the current literature due to the use of different ART methods and the number of methods used during the time allowed (12 months for our study), as well as the different time limits granted according to the studies. Moreover, most cases are described without prior surgical laparoscopy. Finally, it is noteworthy that of the 24 patients who did not obtain pregnancy by ART within 12 months post-ART, six had a pregnancy by ART after >12 months of attempts.

Concerning ART and the chances of achieving a pregnancy after surgery for endometriosis, a recent literature review and meta-analysis demonstrated the same chance of achieving clinical pregnancy and live birth as do women with other causes of infertility [14]. Conversely, there is some consensus concerning fertility outcome after endometriosis surgery, including spontaneous pregnancies [13]. Looking for the mechanisms linking endometriosis to infertility, several mechanisms can be advocated. However, the multifactorial nature of infertility coupled to the variable appearance of endometriosis, make the prognosis of this pathology very challenging. This is attested by the difficulty of some of the current classification for endometriosis in predicting clinical outcomes, including fertility. Regarding the predictivity of the endometriosis staging scores evaluated, our study could not demonstrate the predictive role of the #Enzian score for pregnancy following diagnostic or operative laparoscopy. Our results show that the presence of endometriosis in compartment F of the #Enzian is positively correlated (*p* = 0.079) to a spontaneous pregnancy. This result can be explained, in part, by the extra-pelvic location of some of these lesions. However, it should be taken with a grain of salt, as it is based on the presence of endometriosis in the F compartment in three out of seven women with a spontaneous pregnancy, this number being very low. In addition, the presence of endometriosis in the ureter for one of the three patients should rather demonstrate severe endometriosis. This trend should therefore be verified with a larger sample.

Due to the novelty of #Enzian, we found no study evaluating its prognostic value concerning female fertility and pregnancy rate. Similarly, there are few data concerning the non-revised Enzian classification, as this previous classification was not conceived for studying this outcome [4]. A recent study found a positive correlation of fertility outcome (live birth after IVF—in-vitro fertilization) with compartment A and a negative correlation with the involvement of compartment B and the presence of adenomyosis for the non-revised Enzian classification [15]. However, the authors note that the solidity of results was limited by the small number included in each sub-group of patients included. In order to overcome this limitation, we proposed in our study a pooling of all the scores of the #Enzian compartments in the form of a median of scores. Although this result is not significant, it is however interesting to note that this is the first to propose a pooling of all the scores of the #Enzian compartments in the form of a median of scores. Indeed, the classification in its current form has the drawback of dispersing each case in several categories, making it necessary to improve the number of patients included in a study in order to achieve the necessary statistical power. Future studies could be designed to verify these results.

With regard to the rASRM classification, we found a non-significantly (*p* = 0.054) lower prevalence of pregnancies among stages III–IV, and no other significant association with the other stages and with the mean of the sum of the stages of this classification. The association between infertility and higher rASRM stages is not surprising, even though this difference was not significant. Zeng et al. found similar results, showing a significant difference between stage IV rASRM and lower grades in cumulative pregnancy rate after 1 year. However, the overall data showed that rASRM does not predict endometriosis-associated infertility [16]. A meta-analysis performed in 2014 also found a poor correlation of the rASRM classification with the fertility outcome in terms of live birth rate [14].

Finally, in our study, the EFI index was a score significantly associated with the pregnancy outcome, whether for cumulative pregnancies, those only by ART, or only spontaneous ones. A recent systematic review and meta-analysis confirmed the value of the EFI index in predicting non-ART pregnancies [17]. On the other hand, there is still a lack of evidence as to its reliability for pregnancies after ART [4].

## 5. Strengths and Limitations of the Study

One of the major strengths of this study is that we provided a phone-call follow-up and contacted each patient in order to clarify, if necessary, the anamnesis on the pregnancy, thus making it possible to obtain a more credible outcome. In addition, the study being monocentric allowed the application of the same operating and ART protocols for each patient and thus a standard of treatment. Concerning the limitations, the sample size restricts the possibilities of significant results and of drawing sufficiently valid conclusions. In addition, the retrospective nature of our study limits the quality of some information obtained. This includes the perioperative data acquired (such as the size of the endometriotic lesions or implants, or the extent of the adhesions), influencing the reliability of the calculation of the various scores.

## 6. Conclusions

Although the #Enzian classification was developed with the intent to improve the assessment of female reproductive organs, in our preliminary assessment, we found no association with pregnancy achievement. Meanwhile, the only score significantly associated with the pregnancy outcome in women affected by endometriosis is the EFI score. Additional and larger-scale studies should be performed in order to be able to draw valid conclusions.

## Figures and Tables

**Figure 1 jpm-12-01623-f001:**
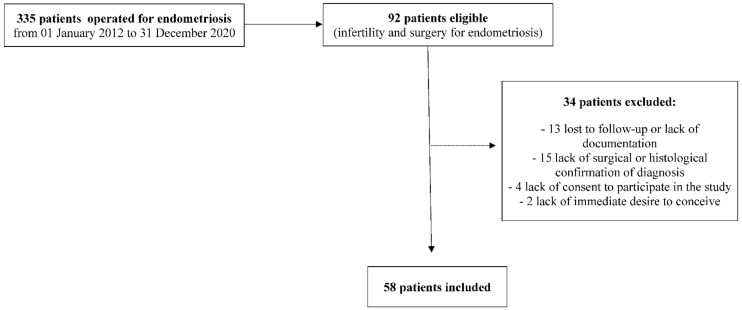
Study population.

**Table 1 jpm-12-01623-t001:** All pregnancies: score subdivided in accordance with the pregnancy outcome.

	No Pregnancy (28)	Pregnancy (30)	*p*
rASRM			
Stage I	46.43% (13/28)	46.67% (14/30)	0.139
Stage II	14.29% (4/28)	36.67% (11/30)	
Stage III	14.29% (4/28)	6.67% (2/30)	
Stage IV	25.00% (7/28)	10.00% (3/30)	
rASRM stage	2.00 (1.00–3.25)	2.00 (1.00–2.00)	0.387
EFI score	7.00 (6.00–8.00)	8.00 (8.00–9.00)	<0.05
Enzian classification			
Peritoneum (P)			
0	21.43% (6/28)	10.34% (3/29)	0.391
1	50.00% (14/28)	51.72% (15/29)	
2	25.00% (7/28)	37.93% (11/29)	
3	3.57% (1/28)	0.00% (0/29)	
Ovary (O) left			
0	67.86% (19/28)	90.00% (27/30)	0.116
1	14.29% (4/28)	3.33% (1/30)	
2	14.29% (4/28)	3.33% (1/30)	
3	0.00% (0/28)	0.00% (0/30)	
Ovary missing	3.57% (1/28)	3.33% (1/30)	
Ovary (O) right			
0	89.29% (25/28)	80.00% (24/30)	0.426
1	3.57% (1/28)	13.33% (4/30)	
2	7.14% (2/28)	6.67% (2/30)	
3	0.00% (0/28)	0.00% (0/28)	
Tube (T) left			
0	88.00% (22/25)	89.3% (25/28)	1.000
1	0.00% (0/25)	0.00% (0/28)	
2	0.00% (0/25)	0.00% (0/28)	
3	0.00% (0/25)	0.00% (0/28)	
Missing	8.00% (2/25)	7.10% (2/28)	
Not visible	4.00% (1/25)	3.60% (1/28)	
Tube (T) left—patency test			
Test not available	10.71% (3/28)	6.67% (2/30)	0.332
Negative test	21.43% (6/28)	6.67% (2/30)	
Positive test	57.14% (16/28)	76.67% (23/30)	
Missing tube	7.14% (2/28)	6.67% (2/30)	
Tube not visible	3.57% (1/28)	3.33% (1/30)	
Tube (T) right			
0	80.77% (21/26)	89.29% (25/28)	0.700
1	0.00% (0/26)	0.00% (0/28)	
2	0.00% (0/26)	0.00% (0/28)	
3	0.00% (0/26)	0.00% (0/28)	
Missing	15.38% (4/26)	7.14% (2/28)	
Not visible	3.85% (1/26)	3.57% (1/28)	
Tube (T) right—patency test			
Test not available	7.14% (2/28)	6.67% (2/30)	0.744
Negative test	21.43% (6/28)	13.33% (4/30)	
Positive test	53.57% (15/28)	70.00% (21/30)	
Missing tube	14.29% (4/28)	6.67% (2/30)	
Tube not visible	3.57% (1/28)	3.33% (1/30)	
Deep endometriosis			
Compartment A			
0	89.29% (25/28)	93.33% (28/30)	0.799
1	3.57% (1/28)	3.33% (1/30)	
2	7.14% (2/28)	3.33% (1/30)	
3	0.00% (0/28)	0.00% (0/30)	
Compartment B left			
0	67.86% (19/28)	56.67% (17/30)	0.700
1	17.86% (5/28)	26.67% (8/30)	
2	14.29% (4/28)	16.67% (5/30)	
3	0.00% (0/28)	0.00% (0/30)	
Compartment B right			
0	71.43% (20/28)	70.00% (21/30)	1.000
1	25.00% (7/28)	23.33% (7/30)	
2	3.57% (1/28)	6.67% (2/30)	
3	0.00% (0/28)	0.00% (0/30)	
Compartment C			
0	100.00% (28/28)	100.00% (29/29)	1.000
1	0.00% (0/28)	0.00% (0/29)	
2	0.00% (0/28)	0.00% (0/29)	
3	0.00% (0/28)	0.00% (0/29)	
Compartment F			
No F locations	89.29% (25/28)	83.33% (25/30)	1.000
Cervix	0.00% (0/28)	3.33% (1/30)	
Colon	3.57% (1/28)	0.00% (0/30)	
Diaphragm	0.00% (0/28)	3.33% (1/30)	
Umbilical	0.00% (0/28)	3.33% (1/30)	
Ureter	7.14% (2/28)	6.67% (2/30)	
Enzian (scores sum)	2.50 (1.00–4.00)	3.00 (2.00–3.00)	0.854
Enzian (scores sum for only P, O, and T)	1.00 (1.00–3.00)	1.50 (1.00–2.00)	0.856

**Table 2 jpm-12-01623-t002:** Pregnancies after ART (assisted reproductive technology): score subdivided in accordance with the pregnancy outcome.

	No Pregnancy (28)	Pregnancy (24)	*p*
rASRM			
Stage I	46.43% (13/28)	41.67% (10/24)	0.125
Stage II	14.29% (4/28)	41.67% (10/24)	
Stage III	14.29% (4/28)	4.17% (1/24)	
Stage IV	25.00% (7/28)	12.50% (3/24)	
rASRM stage	2.00 (1.00–3.25)	2.00 (1.00–2.00)	0.585
EFI score	7.00 (6.00–8.00)	8.00 (7.75–9.00)	<0.05
Enzian classification			
Peritoneum (P)			
0	21.43% (6/28)	12.50% (3/24)	0.489
1	50.00% (14/28)	45.83% (11/24)	
2	25.00% (7/28)	41.67% (10/24)	
3	3.57% (1/28)	0.00% (0/24)	
Ovary (O) left			
0	67.86% (19/28)	87.50% (21/24)	0.402
1	14.29% (4/28)	4.17% (1/24)	
2	14.29% (4/28)	4.17% (1/24)	
3	0.00% (0/28)	0.00% (0/24)	
Ovary missing	3.57% (1/28)	4.17% (1/24)	
Ovary (O) right			
0	89.29% (25/28)	79.17% (19/24)	0.367
1	3.57% (1/28)	16.67% (4/24)	
2	7.14% (2/28)	4.17% (1/24)	
3	0.00% (0/28)	0.00% (0/24)	
Tube (T) left			
0	88.00% (22/25)	86.36% (19/22)	1.000
1	0.00% (0/25)	0.00% (0/22)	
2	0.00% (0/25)	0.00% (0/22)	
3	0.00% (0/25)	0.00% (0/22)	
Missing	8.00% (2/25)	9.09% (2/22)	
Not visible	4.00% (1/25)	4.55% (1/22)	
Tube (T) left—patency test			
Test not available	10.71% (3/28)	8.33% (2/24)	0.431
Negative test	21.43% (6/28)	4.17% (1/24)	
Positive test	57.14% (16/28)	75.00% (18/24)	
Missing tube	7.14% (2/28)	8.33% (2/24)	
Tube not visible	3.57% (1/28)	4.17% (1/24)	
Tube (T) right			
0	80.77% (21/26)	86.36% (19/22)	0.834
1	0.00% (0/26)	0.00% (0/22)	
2	0.00% (0/26)	0.00% (0/22)	
3	0.00% (0/26)	0.00% (0/22)	
Missing	15.38% (4/26)	9.09% (2/22)	
Not visible	3.85% (1/26)	4.55% (1/22)	
Tube (T) right—patency test			
Test not available	7.14% (2/28)	8.33% (2/24)	0.885
Negative test	21.43% (6/28)	12.50% (3/24)	
Positive test	53.57% (15/28)	66.67% (16/24)	
Missing tube	14.29% (4/28)	8.33% (2/24)	
Tube not visible	3.57% (1/28)	4.17% (1/24)	
Deep endometriosis			
Compartment A			
0	89.29% (25/28)	91.67% (22/24)	1.000
1	3.57% (1/28)	4.17% (1/24)	
2	7.14% (2/28)	4.17% (1/24)	
3	0.00% (0/28)	0.00% (0/24)	
Compartment B left			
0	67.86% (19/28)	58.33% (14/24)	0.724
1	17.86% (5/28)	25.00% (6/24)	
2	14.29% (4/28)	16.67% (4/24)	
3	0.00% (0/28)	0.00% (0/24)	
Compartment B right			
0	71.43% (20/28)	66.67% (16/24)	0.896
1	25.00% (7/28)	25.00% (6/24)	
2	3.57% (1/28)	8.33% (2/24)	
3	0.00% (0/28)	0.00% (0/24)	
Compartment C			
0	100.00% (28/28)	100.00% (23/23)	1.000
1	0.00% (0/28)	0.00% (0/24)	
2	0.00% (0/28)	0.00% (0/24)	
3	0.00% (0/28)	0.00% (0/24)	
Compartment F			
No F locations	89.29% (25/28)	91.67% (22/24)	0.887
Cervix	0.00% (0/28)	0.00% (0/24)	
Colon	3.57% (1/28)	0.00% (0/24)	
Diaphragm	0.00% (0/28)	0.00% (0/24)	
Umbilical	0.00% (0/28)	4.17% (1/24)	
Ureter	7.14% (2/28)	4.17% (1/24)	
Enzian (scores sum)	2.50 (1.00–4.00)	3.00 (2.00–3.00)	0.591
Enzian (scores sum for only P, O, and T)	1.00 (1.00–3.00)	2.00 (1.00–2.00)	0.954

**Table 3 jpm-12-01623-t003:** Spontaneous pregnancies: score subdivided in accordance with the pregnancy outcome.

	No Pregnancy (28)	Pregnancy (7)	*p*
rASRM			
Stage I	46.43% (13/28)	71.43% (5/7)	0.587
Stage II	14.29% (4/28)	14.29% (1/7)	
Stage III	14.29% (4/28)	14.29% (1/7)	
Stage IV	25.00% (7/28)	0.00% (0/7)	
rASRM stage	2.00 (1.00–3.25)	1.00 (1.00–1.50)	0.166
EFI score	7.00 (6.00–8.00)	9.00 (8.00–9.00)	<0.05
Enzian classification			
Peritoneum (P)			
0	21.43% (6/28)	0.00% (0/6)	0.746
1	50.00% (14/28)	66.67% (4/6)	
2	25.00% (7/28)	33.33% (2/6)	
3	3.57% (1/28)	0.00% (0/6)	
Ovary (O) left			
0	67.86% (19/28)	100.00% (7/7)	0.570
1	14.29% (4/28)	0.00% (0/7)	
2	14.29% (4/28)	0.00% (0/7)	
3	0.00% (0/28)	0.00% (0/7)	
Ovary missing	3.57% (1/28)	0.00% (0/7)	
Ovary (O) right			
0	89.29% (25/28)	85.71% (6/7)	0.609
1	3.57% (1/28)	0.00% (0/7)	
2	7.14% (2/28)	14.29% (1/7)	
3	0.00% (0/28)	0.00% (0/7)	
Tube (T) left			
0	88.00% (22/25)	100.00% (7/7)	1.000
1	0.00% (0/25)	0.00% (0/7)	
2	0.00% (0/25)	0.00% (0/7)	
3	0.00% (0/25)	0.00% (0/7)	
Missing	8.00% (2/25)	9.09% (0/7)	
Not visible	4.00% (1/25)	4.55% (0/7)	
Tube (T) left—patency test			
Test not available	10.71% (3/28)	0.00% (0/7)	0.835
Negative test	21.43% (6/28)	14.29% (1/7)	
Positive test	57.14% (16/28)	85.71% (6/7)	
Missing tube	7.14% (2/28)	0.00% (0/7)	
Tube not visible	3.57% (1/28)	0.00% (0/7)	
Tube (T) right			
0	80.77% (21/26)	100.00% (7/7)	0.647
1	0.00% (0/26)	0.00% (0/7)	
2	0.00% (0/26)	0.00% (0/7)	
3	0.00% (0/26)	0.00% (0/7)	
Missing	15.38% (4/26)	9.09% (0/7)	
Not visible	3.85% (1/26)	4.55% (0/7)	
Tube (T) right—patency test			
Test not available	7.14% (2/28)	0.00% (0/7)	0.777
Negative test	21.43% (6/28)	14.29% (1/7)	
Positive test	53.57% (15/28)	85.71% (6/7)	
Missing tube	14.29% (4/28)	0.00% (0/7)	
Tube not visible	3.57% (1/28)	0.00% (0/7)	
Deep endometriosis			
Compartment A			
0	89.29% (25/28)	100.00% (7/7)	1.000
1	3.57% (1/28)	0.00% (0/7)	
2	7.14% (2/28)	0.00% (0/7)	
3	0.00% (0/28)	0.00% (0/7)	
Compartment B left			
0	67.86% (19/28)	57.14% (4/7)	0.825
1	17.86% (5/28)	28.57% (2/7)	
2	14.29% (4/28)	14.29% (1/7)	
3	0.00% (0/28)	0.00% (0/7)	
Compartment B right			
0	71.43% (20/28)	85.71% (6/7)	1.000
1	25.00% (7/28)	14.29% (1/7)	
2	3.57% (1/28)	0.00% (0/7)	
3	0.00% (0/28)	0.00% (0/7)	
Compartment C			
0	100.00% (28/28)	100.00% (7/7)	1.000
1	0.00% (0/28)	0.00% (0/7)	
2	0.00% (0/28)	0.00% (0/7)	
3	0.00% (0/28)	0.00% (0/7)	
Compartment F			
No F locations	89.29% (25/28)	57.14% (4/7)	0.079
Cervix	0.00% (0/28)	14.29% (1/7)	
Colon	3.57% (1/28)	0.00% (0/7)	
Diaphragm	0.00% (0/28)	14.29% (1/7)	
Umbilical	0.00% (0/28)	0.00% (0/7)	
Ureter	7.14% (2/28)	14.29% (1/7)	
Any Compartment F location	10.71% (3/28)	42.86% (3/7)	0.079
Enzian (scores sum)	2.50 (1.00–4.00)	2.00 (1.00–2.50)	0.420
Enzian (scores sum for only P, O, and T)	1.00 (1.00–3.00)	1.00 (1.00–2.00)	0.633

## Data Availability

The data that support the findings of this study are available, but restrictions apply to the availability of these data, which was used under license for the current study, and so are not publicly available. Data are however available from the authors upon reasonable request and with permission of the Internal Review Board.

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
