# Peer review of "Endometriosis and Infertility: Prognostic Value of #Enzian Classification Compared to rASRM and EFI Score"

_jpm, 2022, doi:10.3390/jpm12101623_

Round 1

Reviewer 1 Report

This study compares common classification scores for endometriosis and prediction of spontaneous or ART associated pregnancy. Results found that only the EFI score was significantly associated with pregnancy outcomes.

While I believe this is a well written manuscript, I do have a few queries

There appeared to be a large difference between pregnancy and live birth rate in the 58 women recruited for this study. Do the authors think this is a specific to women with endometriosis?

Is there any thought as to why the current classification scores aren’t able to predict pregnancy rates? Perhaps the authors could include some information on the multifactorial nature of endometriosis and how this may be reflected in the inability of the current classification scores to predict pregnancy rates, in the discussion.

Could the authors include a comment on how the pregnancy rates from endometriosis patients undergoing ART in this study compare to similar aged couples without endometriosis? Is the pregnancy rate in this cohort similar to the broader population undergoing ART?

Author Response

While I believe this is a well written manuscript, I do have a few queries

There appeared to be a large difference between pregnancy and live birth rate in the 58 women recruited for this study. Do the authors think this is a specific to women with endometriosis?

Thank you very much for the comment. We noticed the results on viable pregnancies among the achieved pregnancies was not clearly reported. We provided it in the section results. Interestingly, there was a large difference between pregnancy and live birth rate in the group of spontaneous pregnancies compared to the ART ones. We provided a commentary on these data in the section “Discussion”.

Is there any thought as to why the current classification scores aren’t able to predict pregnancy rates? Perhaps the authors could include some information on the multifactorial nature of endometriosis and how this may be reflected in the inability of the current classification scores to predict pregnancy rates, in the discussion.

We provided to include in the section “Discussion” some information on the multifactorial nature of endometriosis and how this may be reflected in the inability of the current classification scores to predict pregnancy rates. Thank you for your advice.

Could the authors include a comment on how the pregnancy rates from endometriosis patients undergoing ART in this study compare to similar aged couples without endometriosis? Is the pregnancy rate in this cohort similar to the broader population undergoing ART?

We included in the section “Discussion” some data and some comment concerning the differences in pregnancy rate in this cohort compared to non-endometriosis cohort and to a broader non-selected population undergoing ART (see the meta-analysis of Bafort et al., 2020).

Reviewer 2 Report

Dear Authors, 

It was wonderful to have the chance to review this interesting paper. 

There are a few modifications that could be made in the:

1. In the introduction part - "as well as the destruction of anatomical structures". I think that this could be modified with distortion of anatomical structures. 

2. In Results section. "Had had" is used two times. 

Regarding presenting the results. There is a question regarding the #Enzian classification. If the authors could give some additional information regarding the use of the #Enzian classification it would be wonderful. The authors stated that the cases included in the study were from January 2012 until December 2020. The #Enzian classification was introduced in January 2021. That means that surgical reports did not include the #Enzian classification. 

Thank you !

Author Response

  1. In the introduction part - "as well as the destruction of anatomical structures". I think that this could be modified with distortion of anatomical structures. 

             Thank you for this comment. We provided to change the phrase accordingly.

  1. In Results section. "Had had" is used two times. 

             Thank you for spotting this typo. We provided to correct it accordingly.

Regarding presenting the results. There is a question regarding the #Enzian classification. If the authors could give some additional information regarding the use of the #Enzian classification it would be wonderful. The authors stated that the cases included in the study were from January 2012 until December 2020. The #Enzian classification was introduced in January 2021. That means that surgical reports did not include the #Enzian classification. 

Many thanks for this comment. Indeed, endometriosis appearance at the surgery was evaluated and and classified according to the #Enzian classification retrospectively by one of the authors (AD), with a revision process of a second one in case of incertainty (AFr). We included this clarification in the section “M&M”.

Reviewer 3 Report

Dear authors, your manuscript deals with comparing the predictive validity of the three most utilized classification scores for endometriosis, #Enzian, EFI, and rASRM, in achieving a spontaneous pregnancy or preg-nancy via assisted reproductive technology (ART) after surgery for endometriosis. Therefore monocentric retrospective study was carried ou. No significant differences were observed concerning the #Enzian score. In conclusions, the revised #En-zian score is not correlated with pregnancy achievement; EFI score is the only score significantly as-sociated with the pregnancy outcome in women affected by endometriosis.

The manuscript is detailed, systematically and carefully written, and deserves attention.

I think the authors conducted a thorough and well-structured systematic review with intriguing findings.

Please find my minor comments below:

- what does the abbreviation EFI mean? Please explain every abbreviation at the first appearance in the text.

Author Response

What does the abbreviation EFI mean? Please explain every abbreviation at the first appearance in the text.

Thanks for your comment. We explained the EFI and each abbreviation comparing for the first time in the manuscript